# Geminivirus C4 proteins inhibit GA signaling via prevention of NbGAI degradation, to promote viral infection and symptom development in *N. benthamiana*

**Pengbai Li[1], Liuming Guo[1], Xinyuan Lang[1], Mingjun Li[1], Gentu Wu[1], Rui Wu[1], Lyuxin Wang[1], Meisheng Zhao[1], Ling Qing**[1,2]*

**1** Chongqing Key Laboratory of Plant Disease Biology, College of Plant Protection, Southwest University, Chongqing, People's Republic of China, **2** National Citrus Engineering Research Center, Southwest University, Chongqing, People's Republic of China

* qling@swu.edu.cn

**Data Availability Statement:** All relevant data are within the manuscript and its Supporting Information files.

## Abstract

The phytohormone gibberellin (GA) is a vital plant signaling molecule that regulates plant growth and defense against abiotic and biotic stresses. To date, the molecular mechanism of the plant responses to viral infection mediated by GA is still undetermined. DELLA is a repressor of GA signaling and is recognized by the F-box protein, a component of the SCF$^{SLY1/GID2}$ complex. The recognized DELLA is degraded by the ubiquitin-26S proteasome, leading to the activation of GA signaling. Here, we report that ageratum leaf curl Sichuan virus (ALCScV)-infected *N. benthamiana* plants showed dwarfing symptoms and abnormal flower development. The infection by ALCScV significantly altered the expression of GA pathway-related genes and decreased the content of endogenous GA in *N. benthamiana*. Furthermore, ALCScV-encoded C4 protein interacts with the DELLA protein NbGAI and interferes with the interaction between NbGAI and NbGID2 to prevent the degradation of NbGAI, leading to inhibition of the GA signaling pathway. Silencing of *NbGAI* or exogenous GA$_3$ treatment significantly reduces viral accumulation and disease symptoms in *N. benthamiana* plants. The same results were obtained from experiments with the C4 protein encoded by tobacco curly shoot virus (TbCSV). Therefore, we propose a novel mechanism by which geminivirus C4 proteins control viral infection and disease symptom development by interfering with the GA signaling pathway.

## Author summary

Gibberellins (GAs) are plant hormones essential for many developmental processes in plants. Plant virus infection can induce abnormal flower development and influence the GA pathway, resulting in plant dwarfing symptoms, but the underlying mechanisms are still not well described. Here, we demonstrate that the geminivirus-encoded C4 protein regulates the GA signaling pathway to promote viral accumulation and disease symptom

**Funding:** This work was supported by the Fundamental Research Funds for the Central Universities (Grant No. XDJK2017A006) awarded to L.Q. The funders had no role in study design, data collection and analysis, decision to publish, or preparation of the manuscript.

**Competing interests:** The authors have declared that no competing interests exist.

development. By directly interacting with NbGAI, the C4 protein interferes with the interaction between NbGAI and NbGID2, which inhibits the degradation of NbGAI. As a result, the GA signaling pathway is blocked, and the infected plants display symptoms of typical dwarfing and delayed flowering. Our results reveal a novel mechanism by which geminivirus C4 proteins influence viral pathogenicity by interfering with the GA signaling pathway and provide new insights into the interaction between the virus and host.

## Introduction

Geminiviruses are a class of DNA viruses with circular single-stranded genomes that encode limited numbers of proteins [1–4]. Geminiviruses recruit host factors to ensure their genome replication, virion assembly, movement, insect transmission, and other processes [2]. Plants infected with geminiviruses often develop disease symptoms such as yellow veins, leaf curling, enation, and curly shoots [4–7]. During virus-plant arms races, host plants have evolved specific proteins targeting various viral infection steps. In response, viruses, including geminiviruses, have evolved specific strategies to hijack specific host factors to interfere with host defense responses [8].

C4 proteins (also known as AC4 proteins of bipartite geminiviruses), encoded by monopartite geminiviruses in the genus *Curtovirus*, *Topocuvirus*, *Turncurtovirus*, and *Begomovirus*, are multifunctional proteins [9, 10]. To date, multiple C4 proteins have been reported as disease symptom determinants that can disrupt host cell development to cause symptom formation [11–14]. Mutational analyses of C4/AC4 proteins have shown that mutations can alleviate disease symptoms and suppress virus accumulation in infected plants [15]. Overexpression of *C4/AC4* in plants induces hyperplasia, callus-like tissues, and curly leaves due to changes in cell cycle-related gene expression [13, 16]. It was reported that the C4 protein encoded by tomato leaf curl Yunnan virus (TLCYnV) could interact with NbSKη to reduce its nuclear accumulation. Cytoplasmic NbSKη is less phosphorylated and has a reduced ability to degrade the cell cycle regulator NbCycD1;1 [17]. Sweet potato leaf curl virus (SPLCV) was reported to interact with brassinosteroid-insensitive 2 (AtBIN2) to relocate AtBIN2-interacting proteins, AtBES1/AtBZR1, into the nucleus to induce abnormal plant development via activation of the BR signaling pathway [18]. Recently, C4 has been proven to suppress RNA silencing through its interaction with receptor-like kinase BARELY ANY MERISTEM 1 (BAM1) and BAM2. These interactions inhibit the functions of BAM1 and BAM2 in spreading RNAi signals to adjacent cells [19]. Cotton leaf curl Multan virus (CLCuMuV) C4 could interact with S-adenosyl methionine synthetase (SAMS) to inhibit its ability to suppress transcriptional and post-transcriptional gene silencing, resulting in more substantial CLCuMuV infection [20]. Geminivirus C4 can also regulate systemic viral infection in plants. For example, the introduction of mutations into beet severe curly top virus (BSCTV) C4 resulted in a systemic movement defective virus [21].

Gibberellins (GAs) are plant hormones that play crucial roles in plant growth and development and defense responses against abiotic and biotic stresses [22–24]. GA-defective mutant plants are stunted, and this stunted phenotype can be restored by spraying the plants with exogenous GA. Although GA-insensitive mutant plants also exhibit a dwarfing phenotype, similar to the GA-defective mutant plants, this phenotype is not reversible through the application of exogenous GA [25]. In contrast, mutant plants that produce constitutively activated GA have elongated stems relative to wild-type plants. The phenotype of these plants directly correlated with the GA content. In the past two decades, the main elements involved in the GA

signaling pathway, including GA receptor protein GID1 (GIBBERELLIN INSENSITIVE DWARF1), DELLA (aspartic acid–glutamic acid–leucine–leucine–alanine), F-box protein SLY1 (SLEEPY1), and SNZ (SNEEZY), have been identified through genetic screening of rice and *Arabidopsis* mutant lines [26].

DELLA is a member of the GRAS family and is an inhibitor of plant growth and development, which functions at the nexus in the GA signaling pathway to negatively regulate GA biosynthesis [27, 28]. Similar to other GRAS family proteins, DELLA has a conserved C-terminal GRAS domain, which is necessary for protein-protein interactions and transcriptional regulation, and an N-terminal DELLA domain. Deletion of the DELLA domain affects its binding to the GA receptor GID1 and results in a semidominant GA-insensitive dwarfing phenotype [29]. DELLA proteins of different plant species, including *Arabidopsis*, wheat, corn, rice and barley, are highly conserved. *Arabidopsis* is known to encode five DELLA proteins (e.g., GAI, RGA, RGL1, RGL2 and RGL3) playing slightly different, but redundant, functions in the inhibition of GA-dependent responses [30]. However, tomato has only one DELLA gene, called *PROCERA* (*PRC*) [31].

Based on the transmission mode, GA can be divided into two major groups: the group utilizing the F-box protein-dependent transmission mode and the other group utilizing the F-box protein-independent transmission mode. After receiving the signal, the receptor protein GID1 binds the active GA and interacts with the negative regulatory DELLA to form a GA-GID1-DELLA complex. DELLA is then recognized by the F-box protein in the SCF$^{SLY1/GID2}$ complex and degraded by the ubiquitin-26S proteasome, leading to the removal of the repressive effect of DELLA [32–36]. For those plants that lack functional F-box proteins, a large amount of DELLA will accumulate in cells, leading to abnormal plant growth and development [37].

Ageratum leaf curl Sichuan virus (ALCScV) is a member of the genus *Begomovirus*, family *Geminiviridae*. ALCScV was first reported in *Ageratum conyzoides* in Sichuan Province, China, in 2018 and can cause severe symptoms on multiple plants [7, 38]. Although ALCScV C4 has been discovered as a disease symptom determinant, the molecular mechanism underpinning this action is unclear. In this study, we found determined that upon ALCScV infection, the expression of GA pathway-related genes and the content of endogenous GA were significantly affected. Further analyses revealed that ALCScV C4 could interact with NbGAI *in vitro* and *in vivo*, which thwarts the interaction of NbGAI and NbGID2, thereby preventing degradation of NbGAI. In addition, silencing of *NbGAI* or spraying plants with exogenous GA$_3$ greatly enhanced *N. benthamiana* resistance to ALCScV infection. Meanwhile, tobacco curly shoot virus (TbCSV)-encoded C4 can also interact with NbGAI, and silencing of *NbGAI* expression can inhibit TbCSV infection. In summary, we propose that one of the specific functions of geminivirus C4 proteins is to interfere with the GA signaling pathway to promote virus infection and disease symptom development, a mechanism that has not been reported previously.

## Results

### ALCScV infection affects the expression of GA pathway-associated genes and the biosynthesis of endogenous GA

The ALCScV-infected *N. benthamiana* plants showed strong dwarfing and abnormal flower development, which were not observed in the mock-inoculated plants or the plants inoculated with ALCScV-mC4 (Fig 1A). The results of statistical analysis supported the symptom observations (S1 Fig). Because previous studies have demonstrated that GA can regulate plant height and flower development [39], the effect of ALCScV infection on GA biosynthesis and the

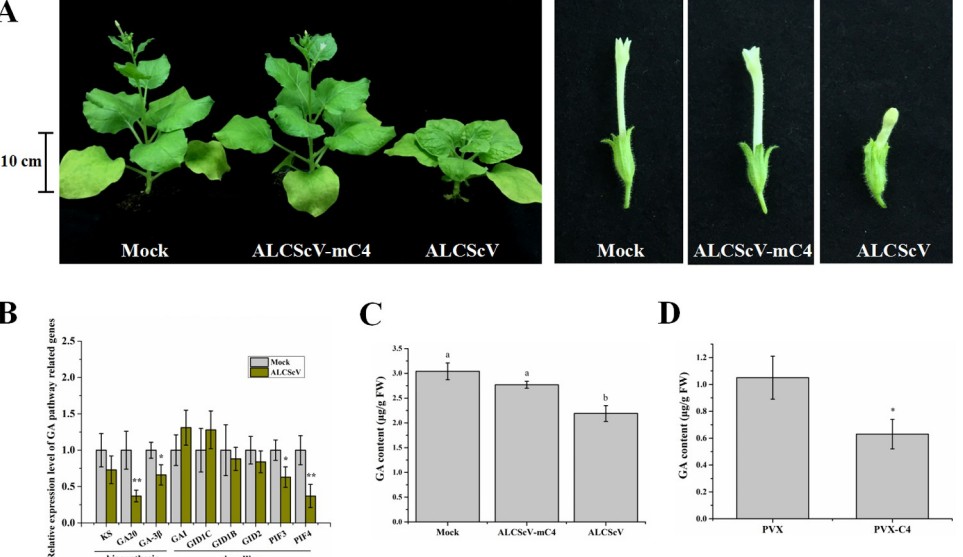

**Fig 1. ALCScV infection influences the gibberellin (GA) pathway in *N. benthamiana* plants.** (A) Photographs of the mock-, ALCScV- and ALCScV-mC4-inoculated *N. benthamiana* plants taken at 28 dpi. The results were reproduced in three independent experiments using 10 plants per treatment. (B) qRT-PCR analyses of the relative expression of GA pathway-associated genes in mock- or ALCScV-inoculated *N. benthamiana* plants. ** indicates a significant difference between the two treatments at the $P < 0.01$ level by Student's $t$-test; * indicates a significant difference between the two treatments at the $P < 0.05$ level. These experiments were performed with three independent biological replicates with similar results. (C) and (D) Results of HPLC analyses showing the contents of endogenous GA in the PVX-, PVX-C4-, mock-, ALCScV-, and ALCScV-mC4-inoculated *N. benthamiana* plants at 14 dpi, respectively. Different letters above the bars indicate significant differences ($P < 0.05$).

expression of GA signaling pathway-associated genes were investigated through qRT-PCR. The results showed that the relative expression levels of *NbGA20*, *NbGA-3β*, *NbPIF3* and *NbPIF4* were significantly down-regulated in the ALCScV-inoculated *N. benthamiana* plants (Fig 1B). Similarly, at 14 days post-agroinfiltration (dpi), the content of endogenous GA was also significantly decreased in the ALCScV-inoculated *N. benthamiana* plant (2.19 µg g$^{-1}$ FW) leaves compared to that in the mock- (3.04 µg g$^{-1}$ FW) or the ALCScV-mC4-inoculated *N. benthamiana* plant (2.77 µg g$^{-1}$ FW) leaves (Fig 1C). In a separate study, the plants inoculated with PVX-C4 also displayed significant dwarfing and delayed flowering (S1 Fig). As expected, the content of endogenous GA in the PVX-C4-inoculated plants (0.63 µg g$^{-1}$ FW) was significantly decreased compared to that in the PVX-inoculated plants (1.05 µg g$^{-1}$ FW) (Fig 1D). Consequently, we speculated that the C4 protein is a regulator of the GA pathway-associated genes as well as the biosynthesis of endogenous GA.

## ALCScV C4 directly interacts with NbGAI

To understand how ALCScV C4 regulates the GA pathway, Y2H assays were performed to screen host protein(s) that could interact with ALCScV C4. The results showed that a GA signaling pathway repressor, NbGAI, interacted with C4 in the yeast cells (Fig 2A). The GAI proteins belong to the family of GRAS transcription factors, of which only one member has been identified in Solanaceae plants [31]. To verify this interaction *in planta*, BiFC assays were performed, and co-expressed of cYFP-C4 and nYFP-NbGAI resulted in a YFP fluorescence signal in both the nucleus and cytoplasm at 48 hours post-incubation (hpi). No YFP fluorescence was observed in the leaves co-infiltrated with cYFP-C4 and nYFP or cYFP and nYFP-NbGAI (Fig 2B). Then the interaction between C4 and NbGAI was further confirmed by a Co-IP assays

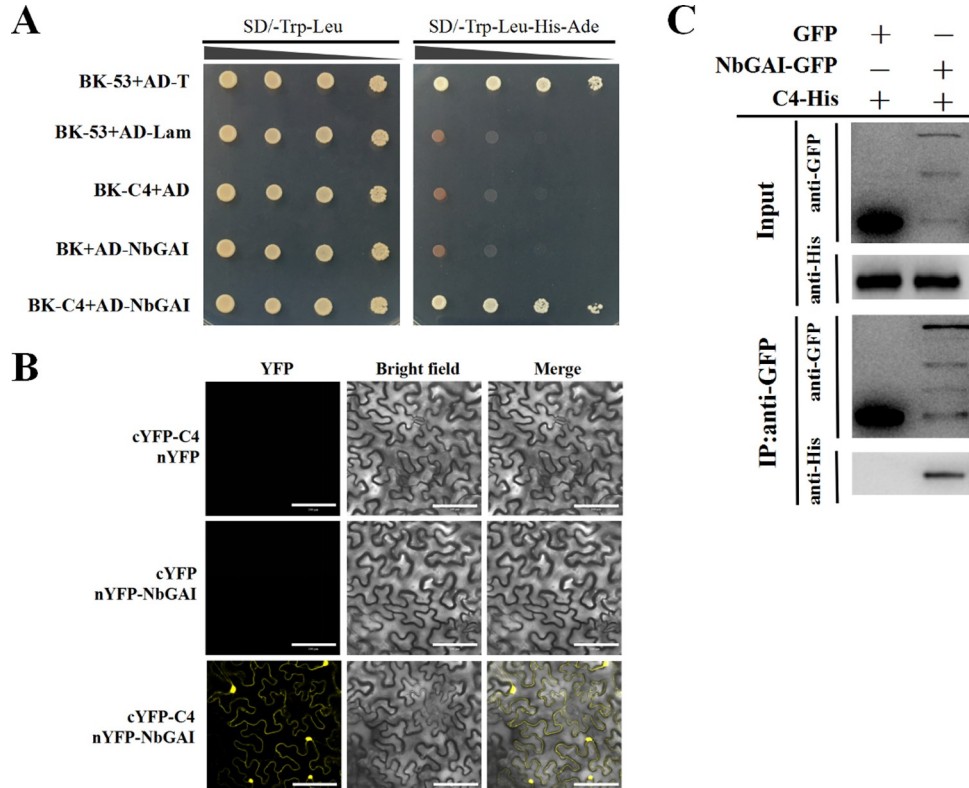

**Fig 2. ALCScV C4 interacts with NbGAI *in vitro* and *in vivo*.** (A) Results of the Y2H assay show the interaction between ALCScV C4 and NbGAI in yeast cells. Recombinant plasmids were co-transformed, in various combinations, into *S. cerevisiae* strain AH109 cells using the lithium acetate method. The transformants were serially diluted 10-fold and then grown on SD/-Trp/-Leu or SD/-Trp/-Leu/-His/-Ade selective medium. Images were taken at 72 hpi. (B) Results of the BiFC assay show the interaction between ALCScV C4 and NbGAI *in vivo*. Leaves co-expressing cYFP-C4 and nYFP, cYFP and nYFP-NbGAI, or cYFP-C4 and nYFP-NbGAI were examined and imaged under a confocal microscope equipped with a FITC filter. Scale bar = 100 *μ*m. (C) Results of the Co-IP assay show the interaction between ALCScV C4 and NbGAI *in vivo*. C4-His and NbGAI-GFP or C4-His and GFP (control) were co-expressed in *N. benthamiana* leaves. Total protein was extracted from the infiltrated leaf samples at 2 dpi. The input and the co-immunoprecipitated proteins were analyzed through Western blot analyses using an anti-GFP or an anti-His antibody.

after transient co-expression of C4-His and NbGAI-GFP in *N. benthamiana* leaves (Fig 2C). Taken together, these results demonstrate that the ALCScV C4 protein directly interacts with the NbGAI protein both *in vitro* and *in vivo*.

## The central region of NbGAI is responsible for its interaction with C4

The coding sequence of *NbGAI* contains 1701 nucleotides and is predicted to encode a hydrophilic protein with 567 amino acids and a molecular mass of approximately 62 kDa (https://web.expasy.org/cgi-bin/protparam/protparam). Amino acid sequence alignment using GAI sequences from *S. lycopersicum* (NP_001234365), *C. baccatum* (PHT30960), *C. maxima* (XP_022998067), *I. triloba* (XP_031108300), *N. benthamiana* (AMO02501), and *N. sylvestris* (XP_009796771) showed that like other GAIs, NbGAI also contains two conserved domains (e.g., the N-terminal DELLA domain and the C-terminal GRAS domain) (S2A Fig). The phylogenetic tree also revealed that NbGAI is more closely related to the GAIs from other Solanaceae plants (S2B Fig).

To explore which region in NbGAI is responsible for the interaction with C4, five mutant plasmids were constructed based on the conserved domain predictions (Fig 3A). The Y2H assays showed that NbGAI-M2 (aa residues 98–566), but not the other four fragments,

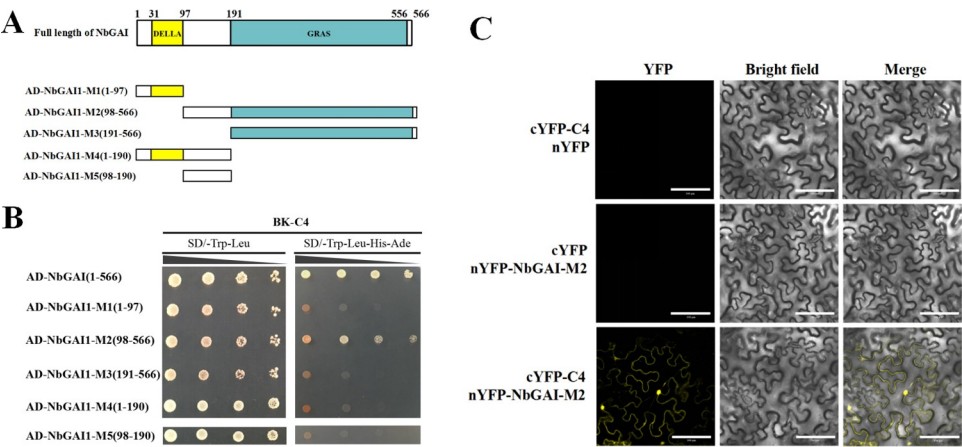

**Fig 3. The middle region of NbGAI is responsible for the interaction with ALCScV C4.** (A) Schematic representations of the NbGAI constructs. The deletion mutants were constructed based on the predictions of the conserved domains in NbGAI. (B) Results of the Y2H assay show the positive interaction between C4 and NbGAI or C4 and NbGAI-M2. (C) Results of the BiFC assay show the interaction between NbGAI-M2 and C4 *in planta*. Scale bar = 100 μm. The procedures of the Y2H and BiFC assays are similar to those described in Fig 2.

interacted with C4 (Fig 3B). This interaction was also confirmed *in planta* through BiFC assays (Fig 3C). Meanwhile, the protein expression of NbGAI-M2 was confirmed by Western blot (S3 Fig). These results suggest that the 98–566 amino acid positions of NbGAI are responsible for the interaction with ALCScV C4.

## The interaction between NbGAI and NbGID2 causes the degradation of NbGAI via the ubiquitin-26S proteasome

Previous studies have shown that the GAI and GID2 interaction results in the degradation of GAI and activation of the GA signaling pathway [22]. In this study, the interaction between NbGAI and NbGID2 was analyzed via Y2H and BiFC assays. The results demonstrated that NbGAI interacted with NbGID2 in yeast cells (Fig 4A) and in *N. benthamiana* leaf cells (Fig 4B).

Because treatment of plants with exogenous $GA_3$ can promote the degradation of GAI via the ubiquitin-26S proteasome [33, 40], we investigated whether NbGAI degradation was also ubiquitin-26S proteasome dependent. cYFP-NbGID2 and nYFP-NbGAI were co-expressed transiently in *N. benthamiana* leaves; at 48 hpi, the green fluorescence signal was observed in the nuclei of the *N. benthamiana* leaf cells (Fig 4C). As expected, after the infiltrated leaves were treated with 50 μM MG132 solution (MG132, a specific inhibitor of the 26S proteasome), the YFP fluorescence intensity in the MG132-treated leaves was significantly stronger than that in the 0.8% ethanol solution (mock)-treated leaves (Fig 4C and 4D). In contrast, the YFP fluorescence intensity in the $GA_3$-treated leaves was significantly decreased compared to the mock-treated leaves (Fig 4C and 4D). Together, these results indicated that the degradation pathway of NbGAI protein in *N. benthamiana* depends on the ubiquitin-26S proteasome.

Further analyses using various NbGAI deletion constructs indicated that the middle region of NbGAI, encompassing amino acids 98–566, is responsible for the interaction between NbGAI and NbGID2 (Fig 4E and 4F).

## ALCScV C4 interferes with the interaction between NbGAI and NbGID2

Because both ALCScV C4 and NbGID2 interacted with NbGAI-M2, but C4 and NbGID2 did not interact with each other (S4 Fig), we speculated that ALCScV C4 and NbGID2 may be

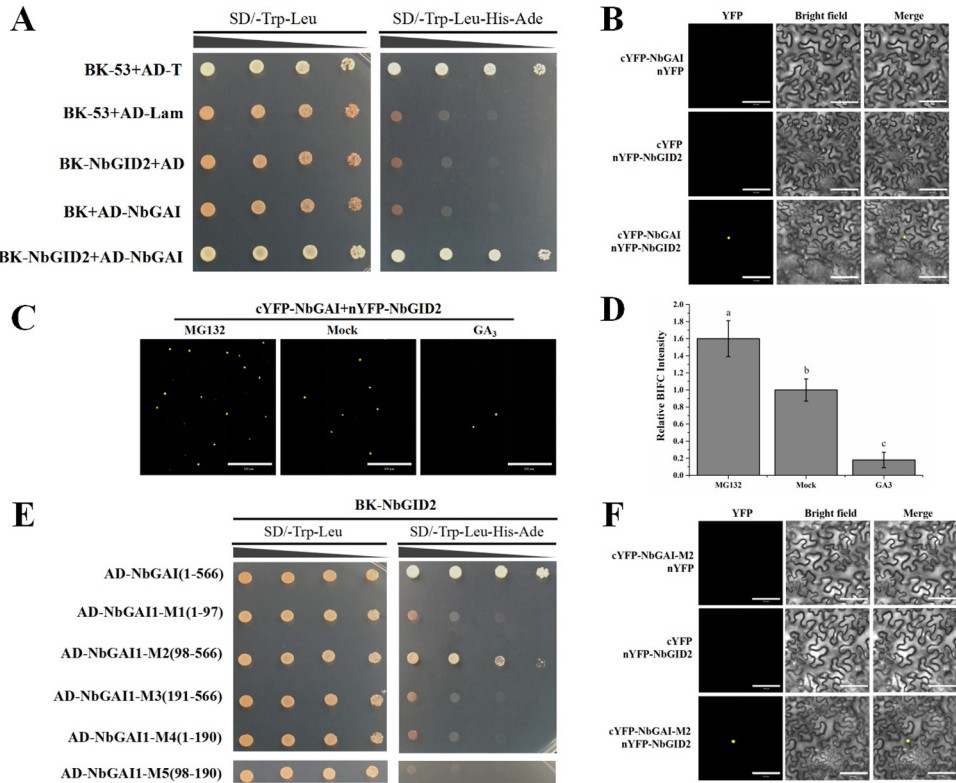

**Fig 4. The interaction between NbGAI and NbGID2 enhances the ubiquitin-26S proteasome-dependent degradation of NbGAI.** (A) Results of the Y2H assay show the interaction between NbGAI and NbGID2. (B) Results of the BiFC assay show the interaction between NbGAI and NbGID2 in *N. benthamiana* leaf cells. Leaves co-expressing cYFP-NbGAI and nYFP, nYFP-NbGID2 and cYFP or cYFP-NbGAI and nYFP-NbGID2 were examined and imaged under a confocal microscope at 48 hpi. Scale bar = 100 $\mu$m. (C) Degradation of NbGAI is ubiquitin-26S proteasome dependent. Fusion cYFP-NbGAI and nYFP-NbGID2 were transiently co-expressed in *N. benthamiana* leaves. The infiltrated leaves were sprayed with a solution containing 50 μM MG132 at 12 h before confocal microscopy or with a solution containing 50 μM GA$_3$ or 0.8% ethanol (control) at 2 h before confocal microscopy. (D) Fluorescence intensities from the BiFC assays were quantified based on the numbers of nuclei with GFP fluorescence. Different letters above the bars indicate significant differences among the treatments at the $P < 0.05$ level. These experiments were performed with three independent biological replicates with similar results. (E) Results of the Y2H assay show the positive interaction between NbGAI-M2 and NbGID2. (F) Results of the BiFC assay show the positive interaction between NbGAI-M2 and NbGID2 *in planta*. Scale bar = 100 $\mu$m. The procedures of the Y2H and BiFC assays are similar to those described in Fig 2.

competitively combining with NbGAI. To test this hypothesis, a competitive BiFC assay was performed. cYFP-NbGID2, nYFP-NbGAI and C4-His or cYFP-NbGID2, nYFP-NbGAI and pCV empty vector were co-expressed in *N. benthamiana* leaves through agroinfiltration. At 48 hpi, the result showed that the YFP fluorescence intensity was significantly decreased in the leaves co-expressing cYFP-NbGID2, nYFP-NbGAI and C4-His compared to that in the leaves co-expressing cYFP-NbGID2, nYFP-NbGAI and pCV empty vector (Fig 5A and 5B). Meanwhile, the Western blot assay confirmed that C4-His was in deed expressed in the infiltrated leaves (Fig 5C). To further verify that ALCScV C4 can interfere with the interaction between NbGAI and NbGID2, a competitive pull-down assay was conducted, and the result indicated that, in the presence of C4-His, the amount of NbGAI-GFP pulled down by NbGID2-Flag was significantly reduced (Fig 5D). In addition, as the amount of C4-His was gradually reduced, the amount of NbGAI-GFP pulled down by NbGID2-Flag was gradually increased, indicating the negative effect of C4 on the interaction between NbGAI and NbGID2.

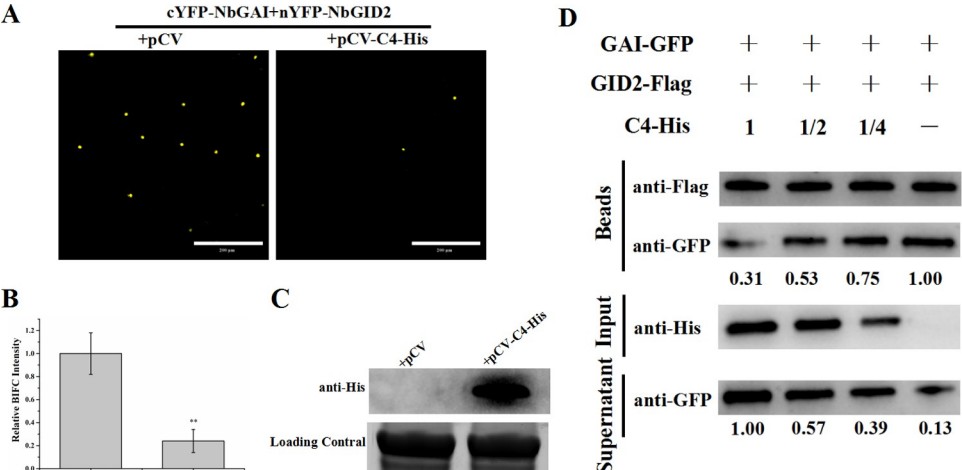

**Fig 5. ALCScV C4 can inhibit the interaction between NbGAI and NbGID2.** (A) Results of the competitive BiFC assay show the inhibitory effect of ALCScV C4 on the interaction between NbGAI and NbGID2. Fusion cYFP-NbGID2, nYFP-NbGAI and C4-His or cYFP-NbGID2, nYFP-NbGAI and empty vector were co-expressed in *N. benthamiana* leaves through agroinfiltration. The infiltrated leaves were harvested at 48 hpi and examined under a confocal microscope. Scale bar = 200 $\mu$m. (B) Results of statistical analyses showing the number of nuclei with YFP fluorescence per examined area. ** indicates a significant difference between the two treatments at the $P < 0.01$ level with Student's *t*-test. These experiments were performed with three independent biological replicates with similar results. (C) Results of the Western blot analysis showing the expression level of ALCScV C4. After electrophoresis, the blot was probed with an anti-His antibody. Coomassie brilliant blue (CBB)-stained RuBisCo large subunit gel to confirm equivalent sample loadings. (D) Results of the competitive pull-down assay show the interaction between NbGAI-GFP and NbGID2-Flag *in vitro*. In this assay, purified NbGAI-GFP (4 μg) was mixed with 0, 2, 4, or 8 μg purified C4-His, respectively, and then incubated with 4 μg immobilized NbGID2-Flag.

## ALCScV C4 can inhibit NbGAI degradation

The interaction between GAI and GID2 can enhance GAI degradation via the ubiquitin-26S proteasome [41]. In this study, to test whether ALCScV C4 could also affect the stability of NbGAI, NbGAI-GFP was transiently expressed in *N. benthamiana* leaves through agro-infiltration, and the green fluorescence from NbGAI-GFP was found predominantly in the nuclei, with a small amount of fluorescence in the cytoplasm at 48 hpi. Meanwhile, the NbGAI-GFP fusion protein was confirmed by the Western blot experiments (S5 Fig). Then, NbGAI-GFP was co-expressed with C4-His or the empty vector in *N. benthamiana* leaf cells, and stronger green fluorescence was observed in the leaves co-expressing NbGAI-GFP and C4-His than in the leaves expressing NbGAI-GFP alone (e.g., NbGAI-GFP and empty vector) (Fig 6A, left two images). However, whether plants were treated with 50 μM GA₃ or MG132, the green fluorescence was enhanced when co-expressed with pCV-C4-His (Fig 6A). Then, the relative accumulation levels of NbGAI-GFP protein were analyzed by Western blot assay. The NbGAI-GFP protein accumulated to significantly higher levels in *N. benthamiana* leaves co-infiltrated with NbGAI-GFP and pCV-C4-His compared to co-infiltration with NbGAI-GFP and pCV empty vector. These results proved that the C4 protein inhibits the degradation of NbGAI (Fig 6B). Furthermore, qRT-PCR and semi-qRT-PCR results indicated that transient expression of C4 did not affect the mRNA expression level of *NbGAI* (Fig 6C and 6D). Meanwhile, *N. benthamiana* expressing GFP alone was used as a control, and the results showed that the C4 protein does not affect the stability of the GFP protein (Fig 6E–6H). In addition, Western blot assays confirmed that the C4 protein did not affect the stability of NbGID2 (S6 Fig). Taken together, these results suggest that the ALCScV C4 protein inhibits the degradation of NbGAI.

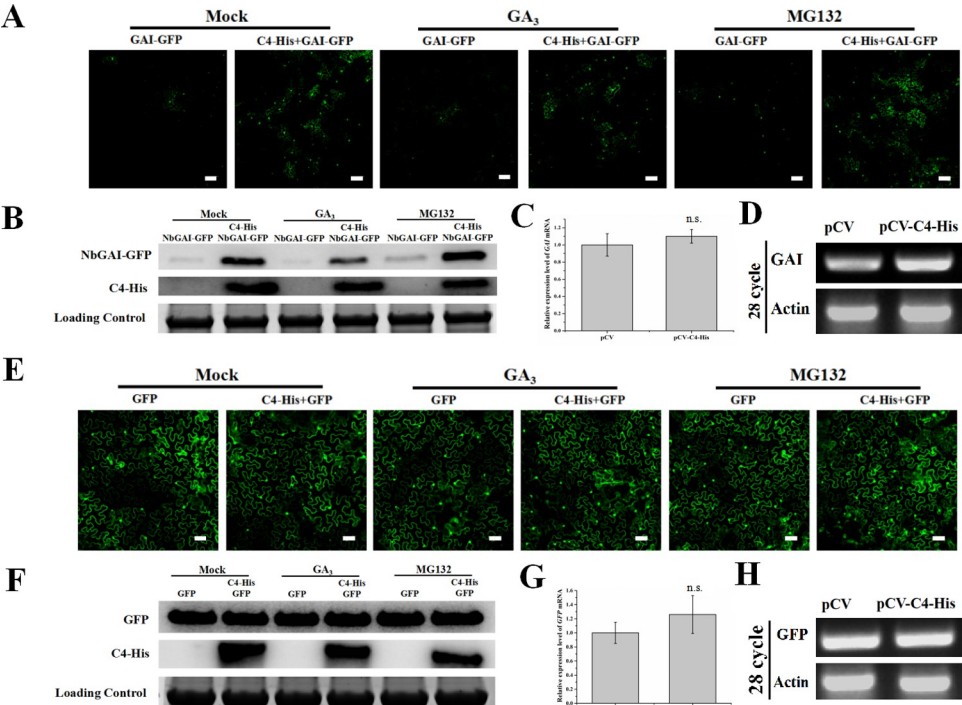

**Fig 6. ALCScV C4 can inhibit NbGAI degradation *in vivo*.** (A) NbGAI-GFP was transiently co-expressed with empty vector or C4-His in *N. benthamiana* leaves. GA₃ (50 μM) and MG132 (50 μM) were applied to the infiltrated leaves 2 h and 12 h before observation, respectively. At 48 hpi, the assayed leaves were harvested and examined under a confocal microscope. Scale bar = 200 $\mu$m. (B) Results of the Western blot analysis show the relative accumulation level of NbGAI-GFP. The blots were probed with an anti-GFP or an anti-His antibody. The CBB-stained RuBisCo large subunit gel shows equivalent sample loadings. (C) and (D) Results of qRT-PCR and semi-qRT-PCR analyses show the relative expression level of *NbGAI*. The expression level of *NbActin* was used as an internal control. The qRT-PCR results were analyzed using the $2^{-\Delta\Delta Ct}$ method. (E) ALCScV C4 does not affect the stability of GFP *in vivo*. Scale bar = 200 $\mu$m. (F) Results of the Western blot analysis show the relative accumulation level of GFP in the mock-, GA₃- or MG132-treated plant leaves. (G) and (H) Results of qRT-PCR and semi-qRT-PCR show the relative expression level of the *GFP* gene. ns indicates not significant according to Student's *t*-test.

To further confirm the ALCScV C4-mediated inhibition of NbGAI degradation, agrobacterium culture carrying pNbGAI-GFP was co-infiltrated with pC4-His, at various concentrations, into *N. benthamiana* leaves. At 2 dpi, Western blot analyses showed that the expression level of NbGAI-GFP increased when the concentration of Agrobacterium culture expressing C4-His was increased, suggesting that the C4-mediated stabilization of NbGAI is C4 dose-dependent (Fig 7A). As expected, C4-His had no clear effect on the expression of GFP (Fig 7B).

## Silencing of *NbGAI* expression in *N. benthamiana* inhibits ALCScV infection

To determine the role of NbGAI in ALCScV infection, *NbGAI* expression was silenced in *N. benthamiana* using a TRV-based VIGS vector and then inoculated with TRV-PDS or TRV-GUS as controls. By 7 dpi, the TRV-PDS-inoculated plants showed photobleaching phenotypes, whereas no obvious symptoms were observed in TRV-GUS-inoculated plants (Fig 8A). As expected, the TRV-NbGAI-inoculated plants resulted in excessive growth compared to TRV-GUS-inoculated plants, which is consistent with a GA-sensitive growth phenotype (Fig 8A). qRT-PCR analysis revealed that the mean transcription level of *NbGAI* in the

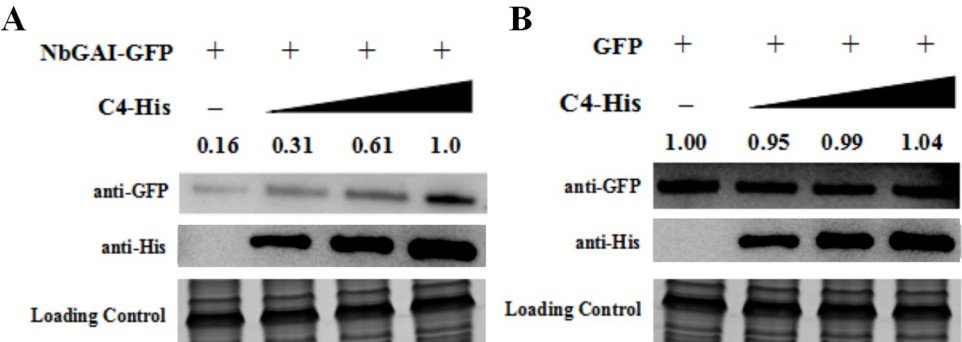

**Fig 7. Inhibition of NbGAI degradation is C4 concentration dependent.** (A) Agrobacterium culture carrying pGAI-GFP and Agrobacterium culture expressing C4-His (diluted to OD600 = 0, 0.25, 0.5 and 1.0, respectively) were co-infiltrated into *N. benthamiana* leaves. Two days later, total protein was extracted from different tissue samples and analyzed through Western blot analyses using an anti-GFP or an anti-His antibody. The CBB-stained RuBisCo large subunit gel shows equivalent sample loadings. (B) Western blot analysis shows that ALCScV C4 does not affect the stability of GFP *in vivo*.

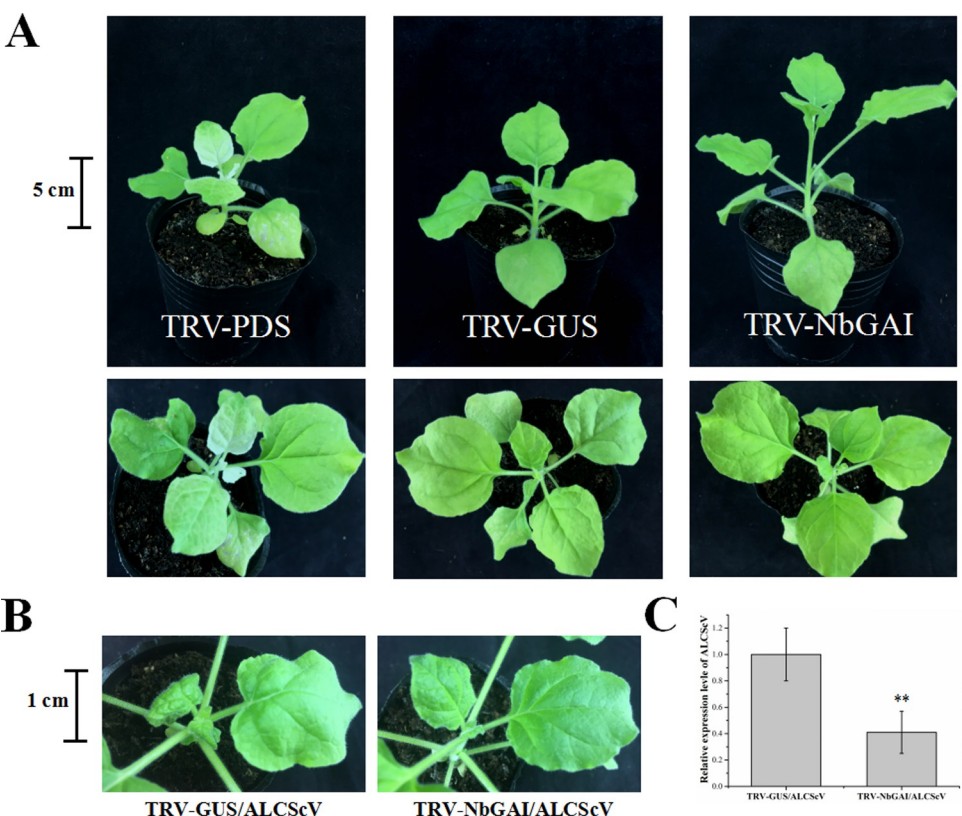

**Fig 8. Silencing of *NbGAI* expression in *N. benthamiana* plants through VIGS inhibits ALCScV infection.** (A) Symptoms of the *N. benthamiana* plants inoculated with TRV-PDS, TRV-GUS or TRV-NbGAI at 7 dpi. (B) The TRV-GUS- and ALCScV-inoculated *N. benthamiana* plants had stronger symptoms of leaf curling and plant dwarfing than the TRV-NbGAI- and ALCScV-inoculated *N. benthamiana* plants at 14 dpi. The results were reproduced in three independent experiments using 10 plants per treatment. (C) The qPCR analysis shows that the accumulation of ALCScV DNA in the TRV-NbGAI- and ALCScV-inoculated plants was significantly reduced compared to the TRV-GUS- and ALCScV-inoculated plants. ** indicates a significant difference between the two treatments at the $P < 0.01$ level. These experiments were performed with three independent biological replicates with similar results.

TRV-NbGAI-inoculated plants was down-regulated by approximately 80% compared to that in the TRV-GUS-inoculated control plants (S7A Fig). In the TRV-NbGAI-inoculated plants, the GA content increased significantly (3.01 μg g$^{-1}$ FW) compared to that in the TRV-GUS-inoculated plants (2.14 μg g$^{-1}$ FW) (S7B Fig). At 14 dpi, the mean height of the TRV-NbGAI-inoculated plants was significantly greater than that of the TRV-GUS-inoculated plants (S7C and S8 Figs). Furthermore, the NbGAI-silenced plants showed a significantly earlier flowering than that of the control plants (S7D and S8 Figs). Then, the effect of NbGAI on ALCScV accumulation and symptom induction was investigated. By 14 dpi, the TRV-GUS- and ALCScV-inoculated *N. benthamiana* plants exhibited severe leaf curling and plant dwarfing symptoms, whereas only mild leaf curling symptoms were observed on the TRV-NbGAI- and ALCScV-inoculated *N. benthamiana* plants (Fig 8B). The qRT-PCR results showed that, compared to the control plants, the accumulation level of ALCScV DNA was significantly reduced in the NbGAI-silenced plants (Fig 8C). Similarly, silencing *NbGAI* can alleviate the ALCScV-induced dwarfing symptoms and promote early flowering (S9 Fig). Taken together, these results indicate that NbGAI plays important roles in viral accumulation and disease symptom development in infected plants.

## Application of exogenous GA$_3$ enhances *N. benthamiana* resistance to ALCScV infection

To investigate how GA$_3$ affects ALCScV infection in plants, *N. benthamiana* plants were inoculated with ALCScV at 12 hours post 50 μM GA$_3$ or 0.8% ethanol treatment (control treatment). By 14 dpi, the control plants showed severe leaf curling and dwarfing symptoms, while the GA$_3$-treated and ALCScV-inoculated plants exhibited only mild leaf curling symptoms (Fig 9A). To evaluate ALCScV accumulation in these plants, the systemic leaf tissues were

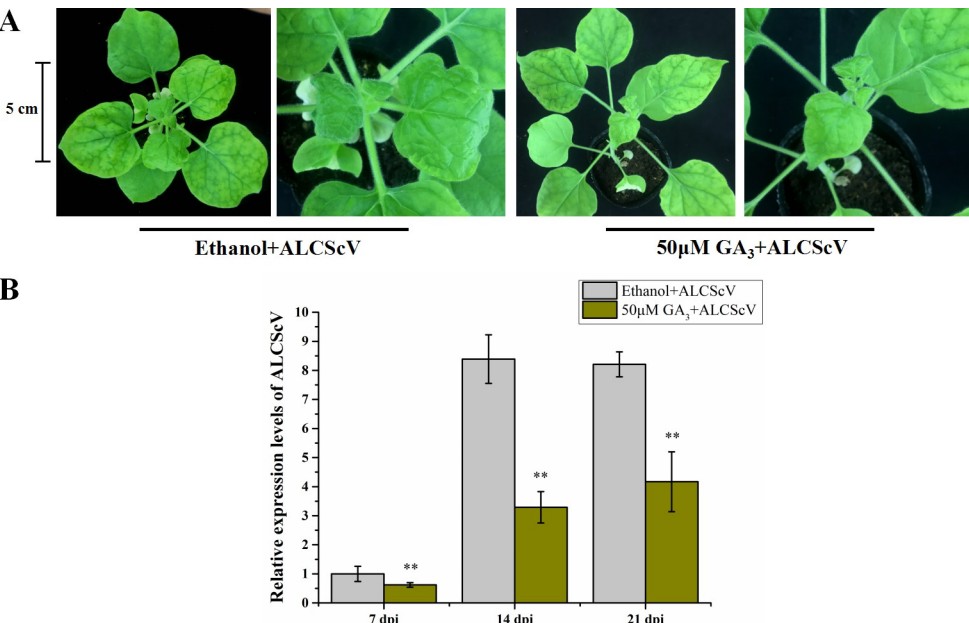

**Fig 9. Application of exogenous GA$_3$ enhances *N. benthamiana* plant resistance to ALCScV infection.** (A) *N. benthamiana* plants were treated with exogenous GA$_3$ or 0.8% ethanol (control) and then inoculated with ALCScV. After ALCScV inoculation, the plants were treated again with GA$_3$ or 0.8% ethanol (once every other day). The plants were photographed at 14 dpi. The results were reproduced in three independent experiments using 10 plants per treatment. (B) The qPCR results show the relative accumulation level of ALCScV DNA in GA$_3$ or 0.8% ethanol-treated plants. $**$ indicates a significant difference between the two treatments at the $P < 0.01$ level with Student's *t*-test. These experiments were performed with three independent biological replicates with similar results.

harvested at 7, 14, and 21 dpi and analyzed individually through qPCR. The results showed that the accumulation level of ALCScV DNA in the GA$_3$-treated and ALCScV-inoculated plants was significantly reduced compared with that in the control plants (Fig 9B). In addition, ALCScV-induced dwarfing symptoms and delayed flowering were significantly reduced by the application of exogenous GA$_3$ (S10 Fig).

## NbGAI can also inhibit the infection of tobacco curly shoot virus (TbCSV) infection in *N. benthamiana*

To further examine whether other begomovirus-encoded C4s could also interact with NbGAI, TbCSV, also a member of the genus *Begomovirus*, was selected as the test virus. Y2H and BiFC assays demonstrated that TbCSV-encoded C4 also interacted with NbGAI and NbGAI-M2 (Figs 10A, 10B and S11). To investigate the effect of NbGAI on TbCSV infection, the NbGAI-silenced (TRV-NbGAI) or non-silenced (TRV-GUS) *N. benthamiana* plants were inoculated with TbCSV. The results showed that the disease symptoms as well as the TbCSV DNA levels in the NbGAI-silenced *N. benthamiana* plants were clearly suppressed compared to those in the non-silenced control plants (Fig 10C and 10D). Similarly, TbCSV C4 could also inhibit the degradation of NbGAI (S12 Fig). Consequently, we propose that begomovirus-encoded C4 proteins are inhibitors of NbGAI degradation and that NbGAI is an important regulator of begomovirus infection and disease symptom development in plants.

## Discussion

As the smallest protein of ALCScV, C4 plays an important role in disease symptom development [38]. To elucidate the molecular mechanism underlying C4-dependent pathogenesis, we

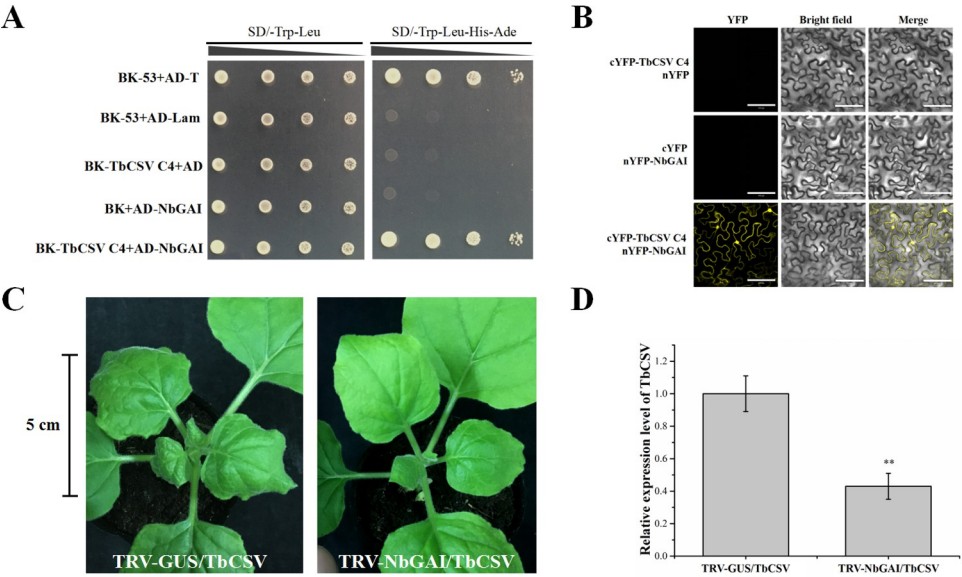

**Fig 10. TbCSV C4 interacts with NbGAI *in vitro* and *in vivo*.** (A) The Y2H assay shows the interaction between TbCSV C4 and NbGAI in yeast cells. (B) The BiFC assay shows the interaction between TbCSV C4 and NbGAI in *N. benthamiana* leaf cells. The agroinfiltrated leaves were harvested at 48 hpi and examined under a confocal microscope equipped with a FITC filter. Bars = 100 $\mu$m. (C) *N. benthamiana* plants were inoculated with TRV-GUS and TbCSV or TRV-NbGAI and TbCSV. The TRV-GUS- and TbCSV-inoculated plants showed severe leaf curling symptoms, while the TRV-NbGAI and TbCSV-inoculated *N. benthamiana* plants showed mild leaf curling symptoms at 7 dpi. The results were reproduced in three independent experiments using 10 plants per treatment. (D) The qPCR analysis shows the accumulation level of TbCSV DNA. ** indicates a significant difference between the two treatments at the $P < 0.01$ level with Student's *t*-test. These experiments were performed with three independent biological replicates with similar results.

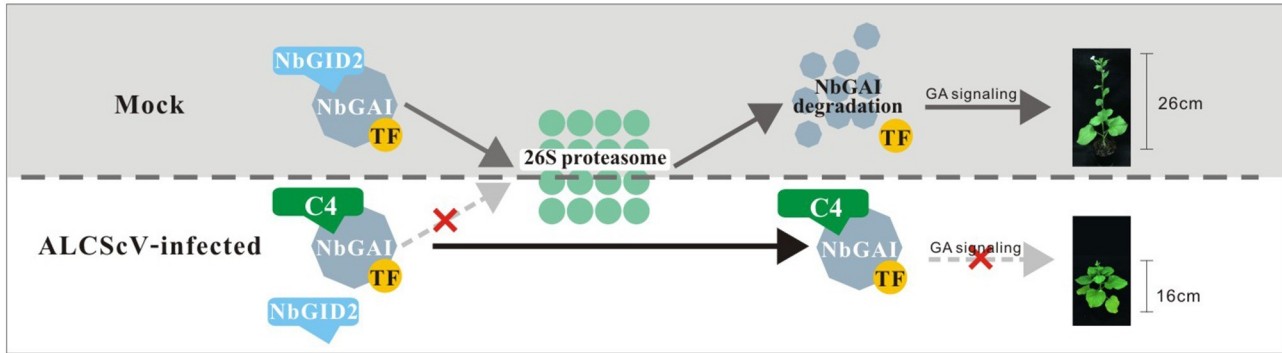

**Fig 11. Proposed model.** In mock *N. benthamiana* plants, the interaction between NbGAI and NbGID2 leads to the ubiquitination and degradation of NbGAI by the 26S proteasome, releasing specific transcription factors. Then, genes of the gibberellin (GA) signaling pathway are adequately regulated by corresponding transcription factors, promoting normal growth of *N. benthamiana* plants. Under ALCScV infection conditions, C4 directly interacts with NbGAI, which interferes with the interaction between NbGAI and NbGID2 to prevent the degradation of NbGAI and block the GA signaling pathway, ultimately leading to symptoms of plant dwarfing and abnormal flower development.

analyzed the function of ALCScV C4 in virus infection and symptom development in *N. benthamiana* and found that ALCScV C4 is a regulator of the GA signaling pathway. Further analyses demonstrated that C4 could directly interact with the negative regulator NbGAI of the GA signaling pathway *in vitro* and *in vivo*, and this interaction prevents the degradation of NbGAI, resulting in a block of the GA signaling pathway, causing infected plants to exhibit typical dwarfing and delayed flowering (Fig 11). Meanwhile, the similar role of the C4 protein encoded by tobacco curly shoot virus (TbCSV), another species of *Begomovirus* [42], was also revealed in this study. It is speculated that the C4-mediated induction of delayed flowering and plant dwarfing via the suppression of the GA signaling pathway are probably common mechanisms among all begomoviruses.

Virus infection in plants causes significant changes in physiological processes, leading to the onset of various disease symptoms [43]. Numerous reports have indicated that GA plays important roles in plant growth and flower development, and plants defective in GA production often show severe plant dwarfing and the development of abnormal flowering [22]. For example, the infection of barley yellow dwarf virus (BYDV) in barley causes a significant reduction in endogenous GA content, leading to plant dwarfing symptoms [44]. Similarly, rice plants infected with rice tungro spherical virus (RTSV) produce significantly less GA, which is considered the main reason for plant dwarfing [45]. Rice dwarf virus (RDV) P2 protein interacted with rice kauri oxidase to suppress GA biosynthesis, which caused rice dwarfing symptoms. In contrast, treatment of rice plants with exogenous GA alleviated rice dwarfing symptoms [39]. In this study, we found that ALCScV infection also causes dwarfing symptoms and delayed flowering, and the infected plants produced significantly less GA. Meanwhile, the ALCScV-induced dwarfing symptoms and delayed flowering were significantly reduced by the application of exogenous $GA_3$, suggesting a direct correlation between dwarfing symptoms and delayed flowering with a reduced endogenous GA content.

DELLA is a key negative regulator of the GA signaling pathway [28]. GA-mediated plant responses are usually regulated by the $SCF^{SLY1/GID2}$ complex, which senses the GA signal and recruits DELLA followed by ubiquitination and then degradation of DELLA through the ubiquitin-26S proteasome, to activate the GA pathway [22, 46]. Sequences of DELLA proteins from different plant species are relatively conserved, suggesting that their functions are similar [47, 48]. DELLA proteins contain two conserved domains. The C-terminal GRAS domain is important for DELLA stability [49]. In this study, we demonstrated that ALCScV C4 can

directly interact with NbGAI, which interferes with the interaction between NbGAI and NbGID2, leading to the prevention of NbGAI degradation and suppression of the GA signaling pathway. Previous studies have demonstrated that to achieve infection in plants, viruses have evolved multiple mechanisms to defeat the plant defense machinery. For example, the RDV P2 protein hijacks OsIAA10 to block the interaction between OsIAA10 and OsTIR1, thereby preventing ubiquitin-26S proteasome-mediated OsIAA10 degradation [50]. Cotton leaf curl Multan betasatellite-encoded βC1 has been proven to interact with NbSKP1, which interferes with the interaction between SKP1 and CUL1 and the function of the SCF complex, resulting in the prevention of GAI degradation and repression of the plant response to GA [51]. Rice black-streaked dwarf virus (RBSDV)-encoded P5-1 can regulate the ubiquitination activity of SCF E3 ligases and inhibit the jasmonate signaling pathway to benefit its infection in rice [52]. Turnip mosaic virus (TuMV) P1 protein interacts with cpSRP54 and mediates its degradation via the 26S proteosome and autophagy pathways, thereby inhibiting the cpSRP54-facilitated delivery of AOCs to the thylakoid membrane and manipulation of JA-mediated defense [53]. Tobacco mosaic virus (TMV) replication-associated protein has been demonstrated to affect the localization and stability of auxin/indole-3-acetic acid (IAA) proteins in *Arabidopsis* to promote its infection [54, 55]. The interaction of cucumber mosaic virus (CMV) 2b protein with JAZ can prevent the recruitment of JAZ by COI1 and the degradation of JAZ, resulting in inhibition of the JA pathway [56]. DELLA not only plays a role directly in the GA signaling pathways but also in other hormone signal transduction pathways in plants. Whether the C4 protein impacts other hormone pathways by inhibiting the degradation of the DELLA protein remains to be further studied.

Previous studies have reported that TbCSV infection can cause a certain degree of dwarf symptoms in plants [6]. In this study, we also found that the TbCSV C4 protein interacts with the NbGAI protein and inhibits its protein degradation, which probably explains why the presence of TbCSV causes a dwarf plant phenotype.

Phytohormones are not only significant signaling molecules that regulate plant growth and development but they also regulate plant abiotic and biotic stress responses [57, 58]. It was reported that auxin signaling plays a positive role in rice defense against RBSDV infection [59]. Application of methyl jasmonate (MeJA) or brassinazole (BRZ) to rice leaves causes significant reductions in RBSDV incidence and virus accumulation [60]. In this study, we found that silencing *NbGAI* expression or applying of exogenous $GA_3$ greatly enhanced *N. benthamiana* resistance to ALCScV infection. Crosstalk between plant hormones is the core of the plant stress response [61]. GA does not work independently but acts in a complex signaling network combined with other plant hormone signaling pathways. For example, application of exogenous GA to *Arabidopsis* plants can increase SA biosynthesis as well as the expression of the *NPR1* gene to enhance plant resistance to abiotic stresses [62]. Navarro et al., found that pretreatment with exogenous GA promotes the resistance of *Arabidopsis* to *Pto* DC3000 by degrading DELLAs and thus changing the balance of SA/JA signaling [63]. It has been shown that DELLAs can interact with BZR1, leading to inhibition of the DNA-binding ability of BZR1, and GA-induced DELLA degradation enhances BR signaling [64, 65]. BR has been shown to play a key role in coping with viral infections. For example, BL (the most active BR) treatment can increase systemic resistance to TMV through the production of reactive oxygen species (ROS) in *N. benthamiana* [66]. Exogenous application of BR has been shown to alleviate tomato yellow leaf curl virus (TYLCV)-induced symptoms in tomato and BCTV C4-induced developmental abnormalities in *Arabidopsis* [67, 68]. In addition, GAs and DELLAs contribute to the fine-tuning of ROS production; among them, GA can induce oxidative stress in plants, lead to programmed cell death, and contribute to enhanced disease resistance

[69]. Therefore, we speculate that GAs influence viral DNA accumulation through crosstalk with other plant endogenous hormones (such as SA, JA, and BR) or regulate the ROS balance.

In summary, the results presented here reveal that geminivirus-encoded C4 proteins can directly interact with NbGAI, which interrupts the interaction between NbGAI and NbGID2, leading to inhibition of NbGAI degradation and blockage of the GA signaling pathway; therefore, the infected plants display symptoms of typical dwarfing and delayed flowering. This is a novel mechanism by which geminivirus C4 proteins influence viral pathogenicity by interfering with the GA signaling pathway.

## Materials and methods

### Plant growth and virus inoculation

*N. benthamiana* plants were grown inside a growth chamber set at 26˚C and 16 h/8 h (light/dark) photoperiod. ageratum leaf curl Sichuan virus (ALCScV), ALCScV-mC4 (e.g., ALCScV lacks the *C4* gene), potato virus X (PVX) and PVX-C4 (PVX with an inserted ALCScV *C4* gene) were from a previously published source and maintained inside the laboratory [37]. These viruses were inoculated to the leaves of 4-6-leaf-old *N. benthamiana* plants though agroinfiltration method.

### Plasmid construction

For yeast two-hybrid (Y2H) assays, the full-length ALCScV *C4* gene sequence was PCR-amplified using gene specific primers (S1 Table), and then cloned into the pGBKT7 vector to produce pBK-C4. The full-length NbGID2, NbGAI, NbGAI deletion derivatives (refer to as NbGAI-M1, NbGAI-M2, NbGAI-M3, NbGAI-M4, and NbGAI-M5, these mutants encode NbGAI amino acid 1–97, 98–566, 191–566, 1–190, and 98–190, respectively) were PCR-amplified and cloned into the pGAD-T7 vector to produce pAD-NbGID2, pAD-NbGAI, pAD-Nb-GAI-M1, pAD-NbGAI-M2, pAD-NbGAI-M3, pAD-NbGAI-M4, and pAD-NbGAI-M5, respectively.

For bimolecular fluorescence complementation (BiFC) assays, the full length sequence of *C4* was amplified using the primer pair cYFP-C4-F/cYFP-C4-R, and then inserted into the pCV-nYFP vector to generate pcYFP-C4. The full-length *NbGAI*, *NbGAI-M2*, and *NbGID2* sequences were also amplified and inserted individually into the pCV-nYFP vector to generate pnYFP-NbGAI, pnYFP-NbGAI-M2, and pcYFP-NbGID2, respectively.

For Co-immunoprecipitation (Co-IP) assays, DNA fragments of C4-His, GFP, and NbGAI-GFP were amplified, individually, double digested with restriction enzymes, and cloned into the pCV vector to generate pC4-His, pGFP, and pNbGAI-GFP, respectively.

To silence *NbGAI* expression in *N. benthamiana*, a fragment of *NbGAI* sequence 437 bp was amplified and cloned into the pTRV2 vector (kindly provided by Professor Fei Yan, Ningbo University, Zhejiang, China) to generate pTRV2-NbGAI. Plasmid pTRV1, pTRV2-GUS, and pTRV2-PDS were also from the same source.

### Yeast two-hybrid assay

The recombinant plasmids were co-transformed into *Saccharomyces cerevisiae* strain AH109 cells as indicated in the legends of Figs 2, 3 and 4 using the lithium acetate method. The transformed cells were grown on the SD/-Trp-Leu medium. After 2–3 days, positive colonies were selected and cultured on the SD/-Leu-His-Trp-Ade medium plate after being diluted to the concentration of $10^{-1}$, $10^{-2}$, $10^{-3}$, and $10^{-4}$, respectively.

## BiFC assay

Plasmid pcYFP-C4 and pnYFP-NbGAI were individually transformed into *Agrobacterium tumefaciens* strain EHA105 cells, and cultured overnight in a LB medium supplemented with kanamycin (50 μg/mL) and rifampicin (20 μg/mL). The cultures were pelleted, resuspended in an induction buffer (10 mM $MgCl_2$, 100 mM MES, pH 5.7, and 2 mM acetosyringone, in distilled water) till $OD_{600}$ = 0.8–1.0 followed a 3 h incubation at room temperature. The two agrobacterium cultures were mixed at a 1:1 (v/v) ratio, and then infiltrated into *N. benthamiana* leaves with needleless syringes. Mixed agrobacterium cultures carrying pcYFP-C4 and pnYFP (empty vector) or pcYFP (empty vector) and pnYFP-NbGAI were served as controls. At 36–48 hpi, the leaves were collected and examined under a confocal microscope (LEICA, Germany) equipped qith an FITC filter.

## Co-IP assay

For Co-IP assays, mixed agrobacterium cultures carrying plasmid pC4-His and pNbGAI-GFP, or pC4-His and pGFP were infiltrated into *N. benthamiana* leaves. At 2 dpi, the infiltrated leaves were harvested for total protein extraction in an extraction buffer (50 mM Tris-HCl, pH 8.0, 0.5 M sucrose, 1 mM $MgCl_2$, 10 mM EDTA, 5 mM dithionthreitol). The resulting protein samples were individually incubated with anti-GFP agarose beads (Sigma) for 4 h at 4˚C followed by five rinses in an ice-cold PBS buffer. The input and co-immunoprecipitated protein samples were analyzed through Western blot assays using an anti-His or an anti-GFP antibody.

## In *vitro* competitive pull-down assay

For competitive pull-down assay in *vitro*, C4-His, NbGAI-GFP, and NbGID2-Flag fusion proteins were purified by in *vitro* translation system (The Wheat Germ Extract Systems), respectively. The purified NbGAI-GFP (4 μg) was mixed with 0, 2, 4, or 8 μg purified C4-His, and then incubated with 4 μg immobilized NbGID2-Flag for 2 h at 4˚C. After centrifugation and several rinses, the bead-bound proteins and the proteins in the supernatants from different treatments were analyzed through Western blot assays using an anti-Flag, anti-His, or an anti-GFP antibody.

## VIGS and transient gene expression assays

For VIGS, plasmid pTRV1, pTRV2, pTRV2-NbGAI, pTRV2-GUS, and pTRV2-PDS were individually transformed into *A. tumefaciens* strain EHA105 cells. After overnight growth in the liquid LB medium, the cultures were pelleted individually, resuspended in an induction buffer followed by 3 h incubation at room temperature. The agrobacterium culture harboring pTRV1 was diluted to $OD_{600}$ = 1.0 and mixed with an equal volume of diluted agrobacterium culture harboring pTRV2 (refers to as TRV), pTRV2-NbGAI (TRV-NbGAI), pTRV2-GUS (TRV-GUS) or pTRV2-PDS (TRV-PDS). The mixed agrobacterium cultures were individually infiltrated into the leaves of *N. benthamiana* plants at the 3-5-leaf-stage using needless syringes. The plants inoculated with TRV-GUS or TRV-PDS were used as the controls. At 7 dpi, systemic leaves of the assayed plants were harvested for total RNA isolation followed by qRT-PCR analyses. The expression level of *NbActin* in individually tissue samples was used as the internal control. This experiment was repeated three times.

For transient gene expression assays, agrobacterium cultures harboring specific expression vectors were individually grown, pelleted and induced as described above. Each agrobacterium culture was diluted to $OD_{600}$ = 1.0 before infiltration into leaves of *N. benthamiana* plants at 4-6-leaf-stage. This experiment was repeated three times with 15 plants per treatment.

## Hormone level analysis

Leaves of the TRV-GUS- and TRV-NbGAI-inoculated *N. benthamiana* plants were collected at 7 dpi, and the leaves of the mock-, ALCScV-, ALCScV-mC4-, PVX-, and PVX-C4-inoculated *N. benthamiana* plants were collected at 14 dpi. The harvested leaf tissues were weighed, stored at -80°C, and analyzed for endogenous hormone contents by Suzhou Grace Biotechnolgy Co., Ltd through high performance liquid chromatography (HPLC).

## GA₃ treatment and ALCScV inoculation

To test the effect of $GA_3$ on ALCScV infection, *N. benthamiana* plants at the 4-6-leaf stage were sprayed with a solution containing 50 μM $GA_3$ or 0.8% ethanol (mock treatment). After 12 h, the sprayed plants were inoculated with ALCScV through agroinfiltration method. The inoculated plants were grown inside the growth chamber and sprayed with the 50 μM $GA_3$ solution or the 0.8% ethanol solution once every other day till 7 dpi. Leaves of these plants were harvested and analyzed for ALCScV accumulation through qPCR assays.

## Protein sequence alignment

Sequences of GAI proteins from *Solanum lycopersicum* (GenBank No. NP_001234883.1), *N. tabacum* (GenBank No. BAG68655.1), *Ipomoea batatas* (GenBank No. XP_031118124.1), *Capsicum annuum* (GenBank No. XP_016540318.1), *Camellia sinensis* (GenBank No. ANB66339.1) and *A. thaliana* (GenBank No. NP_001319041.1) were downloaded from the National Center for Biotechnology Information (https://www.ncbi.nlm.nih.gov/) in the US. Alignment of these sequences was performed using the Clustal V software in the MEGA7.0 and the phylogenetic tree was constructed using the MEGA7.0 softwear. Conserved domains in these GAI proteins are then identified.

## Quantitative reverse transcription polymerase chain reaction (qRT-PCR)

Total RNA was extracted from the harvested leaf tissues using the RNAiso Plus reagent (TaKaRa, Dalian, China). Complementary DNA was synthesized using the RT reagent Kit supplemented with a gDNA Eraser (TaKaRa, Dalian, China) and random primers. Quantitative PCR was conducted using the NovoStart SYBR qPCR SuperMix Plus kit (Novoprotein, Shanghai, China). Relative gene expression levels were calculated using the $2^{-\Delta\Delta Ct}$ method [70]. The expression level of *NbActin* gene was used as an internal control during the assays. For each treatment, three technical replicates were used and each treatment was repeated three times. The final result was presented as the mean of three experiments.

## DNA extraction and viral DNA accumulation analysis

Total DNA was extracted using the CTAB method. The relative accumulation levels of viral DNA were measured using the qPCR assays as described previously [37].

## Western blot analysis

Total protein was extracted from leaf samples in a lysis buffer (Biotime, Shanghai, China), separated in 12% sodium dodecyl sulphate-polyacrylamide (SDS-PAGE) gels through electrophoresis, and transferred to PVDF membranes. The membranes were probed individually with a specific rabbit antibody followed by a horseradish peroxidase (HRP)-conjugated goat anti-rabbit IgG antibody. The detection signal was visualized through incubation of the membranes in an ECL solution.

## Supporting information

**S1 Fig. ALCScV C4 influences plant height and flower development.** (A) The photographs of the mock-, ALCScV-, and ALCScV-mC4-inoculated *N. benthamiana* plants at 28 dpi. (B) Results of statistical analysis show the height of the mock-, ALCScV-, and ALCScV-mC4-inoculated *N. benthamiana* plants at 28 dpi. Different letters above the bars indicate significant differences at the $P < 0.05$ level. (C) Results of statistical analysis show the flowering time of the mock-, ALCScV-, and ALCScV-mC4-inoculated *N. benthamiana* plants. (D) The photographs of the PVX- and PVX-C4-infected *N. benthamiana* plants at 14 dpi. (E) Results of statistical analysis show the height of the PVX- and PVX-C4-inoculated *N. benthamiana* plants at 14 dpi. (F) Results of statistical analysis show the flowering time of the PVX- and PVX-C4-inoculated *N. benthamiana* plants. These experiments were performed with three independent biological replicates with similar results.
(TIF)

**S2 Fig. Sequence alignment and phylogenetic analysis of GAI proteins from different plant species.** (A) Analysis of conserved domains in the GAI proteins from different plant species. The selected plant species and gene accession numbers are *Solanum lycopersicum* (NP_001234365), *Capsicum baccatum* (PHT30960), *Cucurbita maxima* (XP_022998067), *Ipomoea triloba* (XP_031108300), *Nicotiana benthamiana* (AMO02501), and *N. sylvestris* (XP_009796771). (B) Phylogenetic tree was constructed using the GAI protein sequences from *N. benthamiana* (AMO02501), *N. attenuata* (XP_019232709), *N. tomentosiformis* (XP_009590329), *N. sylvestris* (XP_009796771), *S. tuberosum* (NP_001305514), *S. lycopersicum* (NP_001234365), *C. baccatum* (PHT30960), *I. triloba* (XP_031108300), *Sesamum indicum* (XP_011099949), *Actinidia deliciosa* (AHB17746), *A. thaliana* (CAA75492), *Glycine soja* (XP_028212758), *C. maxima* (XP_022998067), *Zea mays* (NP_001306661), and *Oryza sativa* (AAR31213). The red triangle indicates the GAI protein of *N. benthamiana* obtained in this study.
(TIF)

**S3 Fig. Western blot analysis the expression of NbGAI-M2 protein.** The blot were probed with anti-His antibodies, coomassie brilliant blue (CBB) staining of RuBisCo large subunit gel to confirm equivalent sample loadings.
(TIF)

**S4 Fig. ALCScV C4 does not interact with NbGID2.** The recombinant plasmids were co-transformed, in various combinations, into *S. cerevisiae* strain AH109 cells using the lithium acetate method. The transformants were 10-fold serially diluted and then grown on the SD/-Trp/-Leu or the SD/-Trp/-Leu/-His/-Ade medium plates for 3 days. The result show that the C4 does not interact with NbGID2.
(TIF)

**S5 Fig. Analysis of NbGAI subcellular localization.** (A) Leaf tissues expressing GFP (top image) or NbGAI-GFP (bottom image) area shown in the left column. Images in the left column were captured under the UV light. The images in the middle column were captured under the bright field. The images in the right column are merged images. Scale bar = 50 $\mu$m. (B) Result of Western blot analysis shows the expression level of GFP and NbGAI-GFP fusion protein. The blot was probed with an anti-GFP antibody. The CBB-stained RuBisCo large subunit gel to confirm equivalent sample loadings.
(TIF)

**S6 Fig. ALCScV C4 protein did not affect the stability of NbGID2 *in vivo*.** NbGAI-GFP together with NbGID2-Flag were co-expressed with C4-His or the empty vector in *N. benthamiana* leaf cells, at 2 dpi, the relative accumulation levels of NbGAI-GFP and NbGID2-Flag protein were analyzed by Western blot assay.
(TIF)

**S7 Fig. Silencing of *NbGAI* inhibits ALCScV infection.** (A) qRT-PCR analysis the relative mRNA levels of NbGAI gene in *N. benthamiana* plants infected with TRV-GUS and TRV-GAI. At 7 dpi, total RNA were extracted from systemically infected leaves, *NbActin* gene was selected as internal controls for the assays. All the gene expression data were performed with three biological replicates and three technical replicates, and the relative genes expression levels using the $2^{-\triangle\triangle Ct}$ method for analysis. (B) The endogenous gibberellin concentrations of TRV-GUS-inoculated and TRV-GAI-inoculated *N. benthamiana* plants, the leaves were collected at 7 dpi, respectively. (C) Statistical analysis of plant height of TRV-GUS-inoculated and TRV-GAI-inoculated *N. benthamiana* plants at 14 dpi. (D) Statistical analysis of flowering time of TRV-GUS-inoculated and TRV-GAI-inoculated N. benthamiana plants. "**" indicate an extremely significant difference ($P < 0.01$ by the Student's t-test), "*" indicate a significant difference ($P < 0.05$ by the Student's t-test). These experiments were performed with three independent biological replicates with similar results.
(TIF)

**S8 Fig. The photographs of the TRV-GUS- and TRV-NbGAI-inoculated *N. benthamiana* plants at 28 dpi.** The results show that silencing *NbGAI* expression in *N. benthamiana* plants through VIGS significantly increases plant height, but early flowering. The results were reproduced in three independent experiments using 10 plants per treatment.
(TIF)

**S9 Fig. Silencing *NbGAI* can alleviate the ALCScV-induced dwarfing symptom and delayed flowering.** (A) Statistical analysis of plant height of mock-inoculated and ALCScV-inoculated *N. benthamiana* plants at 14 dpi. (B) Statistical analysis of flowering time of mock-inoculated and ALCScV-inoculated *N. benthamiana* plants.
(TIF)

**S10 Fig. Application of exogenous GA$_3$ increases plant height, but early flowering.** (A) The mock- or ALCScV-inoculated *N. benthamiana* plants were treated with GA$_3$ or 0.8% ethanol (control), respectively. The plants were photographed at 14 dpi. The results were reproduced in three independent experiments using 10 plants per treatment. (B) Results of statistical analysis show the height of the mock- and ALCScV-inoculated *N. benthamiana* plants, respectively, at 14 dpi. (C) Results of statistical analysis show the flowering time of the mock- and ALCScV-inoculated *N. benthamiana* plants, respectively. Different letters above the bars indicate the significant differences among the treatments at the $P < 0.05$ level.
(TIF)

**S11 Fig. NbGAI-M2 is responsible for the interaction with TbCSV C4.** (A) Results of the Y2H assay show that NbGAI-M2 is responsible for the interaction with TbCSV C4. (B) Results of the BiFC assay agrees with the Y2H assay result.
(TIF)

**S12 Fig. TbCSV C4 inhibits the degradation of NbGAI.**
(TIF)

**S1 Table. Sequence of primers used in this study.**
(DOCX)

**S2 Table. Statistics to support this study.**
(XLS)

## Acknowledgments

We gratitude Professor Fei Yan (Ningbo University, Zhejiang, China) for providing the expression vector TRV and pCV, and Professor Xueping Zhou (Institute of Plant Protection, Chinese Academy of Agricultural Sciences, Beijing, China) and Xiaofeng Cui (Shanghai Institutes for Biological Sciences, Shanghai, China) for providing suggestions for research idea, and Professor Xinshun Ding (The Samuel Roberts Noble Foundation, USA) and Mengji Cao (Southwest University, Chongqing, China) for revising this manuscript.

## Author Contributions

**Conceptualization:** Ling Qing.

**Data curation:** Pengbai Li, Xinyuan Lang, Mingjun Li, Gentu Wu, Rui Wu, Lyuxin Wang, Meisheng Zhao.

**Formal analysis:** Pengbai Li, Liuming Guo.

**Funding acquisition:** Ling Qing.

**Investigation:** Pengbai Li, Xinyuan Lang, Mingjun Li, Ling Qing.

**Methodology:** Pengbai Li.

**Project administration:** Pengbai Li, Ling Qing.

**Resources:** Ling Qing.

**Supervision:** Pengbai Li, Ling Qing.

**Validation:** Pengbai Li, Xinyuan Lang.

**Visualization:** Pengbai Li, Liuming Guo, Ling Qing.

**Writing – original draft:** Pengbai Li.

**Writing – review & editing:** Pengbai Li, Liuming Guo, Mingjun Li, Gentu Wu, Ling Qing.

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
