## [Decision Letter · Decision Letter 0]

8 Feb 2022

Dear Dr. Qing,

Thank you very much for submitting your manuscript "Geminivirus C4 proteins inhibit GA signaling via prevention of NbGAI degradation, to promote viral infection and symptom development in N. benthamiana" for consideration at PLOS Pathogens. As with all papers reviewed by the journal, your manuscript was reviewed by members of the editorial board and by several independent reviewers. In light of the reviews (below this email), we would like to invite the resubmission of a significantly-revised version that takes into account the reviewers' comments.

We cannot make any decision about publication until we have seen the revised manuscript and your response to the reviewers' comments. Your revised manuscript is also likely to be sent to reviewers for further evaluation.

Sincerely,

Aiming Wang, Ph.D

Associate Editor

PLOS Pathogens

Peter Nagy

Section Editor

PLOS Pathogens

Kasturi Haldar

Editor-in-Chief

PLOS Pathogens

orcid.org/0000-0001-5065-158X

Michael Malim

Editor-in-Chief

PLOS Pathogens

orcid.org/0000-0002-7699-2064

Reviewer's Responses to Questions

**Part I - Summary**

Reviewer #1: In this study, the authors demonstrate that geminiviruses encoded C4 proteins regulates the GA signaling pathway to promote viral accumulation and disease symptom development. They found that the C4 protein interferes with the interaction between NbGAI and NbGID2 to inhibit the degradation of NbGAI, which led to the blocking of the GA signaling pathway and the symptom appearances of infected plants. This findings reveal a novel mechanism by which geminivirus C4 proteins influence viral pathogenicity via interfering the GA signaling pathway, and provide new insights into the interaction between virus and host.

Reviewer #2: The manuscript by Li et al. describes how the C4 protein from the geminivirus ageratum leaf curl Sichuan virus interacts with NbGAI and interferes with its 26S proteasome-mediated degradation, repressing GA signaling, which in turns promotes viral accumulation. The authors also show that the C4 protein encoded by a different geminivirus, tobacco curly shoot virus, exerts a similar effect. Therefore, the C4-mediated interference with GA signaling is proposed to be a new geminiviral virulence strategy.

The work presented in this manuscript is timely, novel, and elegantly designed. Nevertheless, I have a few comments that the authors may consider.

**Part II – Major Issues: Key Experiments Required for Acceptance**

Reviewer #1: 1 The protein expression of GAI-M2 in Fig 3 should be confirmed by western blot;

2 It is known that the interaction between GAI and GID2 can enhance GAI, degradation via the ubiquitin-26S proteasome' In this study, to test whether ALCScV C4 could also affect the stability of NbGAI, NbGAI-GFP was transiently expressed in N. benthamiana leaves through agro-infiltration,NbGAI-GFP was co-expressed with C4-His or the empty vector in N. benthamiana leaf cells. Authors should analyze 'NbGAI-GFP together with GID2-tag was co-expressed with C4-His or the empty vector in N. benthamiana leaf cells', to confirm whether ALCScV C4 could affect the stability of NbGAI and GID2-

3 GAs are essential for many developmental processes in plants. Therefore, it is reasonable that C4 interferes with GAI to induce abnormal development. However, how GAs influence viral DNA accumulations? Which resistance pathway is involved in this? The authors should discuss it.

Reviewer #2: Major comments:

- Biological replicates are needed – and should be indicated – for all experiments. Having three plants per experiment (as in the case of the exogenous GA treatment presented in Figure 10) is not sufficient and does not provide an idea of the reproducibility of the results.

- The quality of the confocal images is in general not sufficient.

- English needs editing throughout the manuscript.

- Multiple references are missing in the introduction. The authors can check, for example, the recent review by Devendran et al. (2022) on geminivirus-encoded proteins, or the C4-specific review by Medina-Puche et al. (2021).

**Part III – Minor Issues: Editorial and Data Presentation Modifications**

Reviewer #1: 1 Lack scale bars in the picture with plants, Fig 1A, 8A, 8F,9A, 10C

2 Scale bars in con-focal images are too small and too obscure to read;

3 Clear and amplified con-focal images should be replaced with the current images, which are hardly to see the details of their subcellular localisations in the cell;

4 Fig 8-B-C-D-E should be incorporated into one figure;

5 There should be a space between number and dpi;

6 Line 483 Rice Line 83, 506 Cotton.

Reviewer #2: Minor comments:

- In Figure 10, how was the treatment performed? Repeatedly, or only once? The effect of the hormone treatment of the ability of Agrobacterium tumefaciens to transform plant cells should be tested.

- The authors should discuss the GA/BR interplay. This is particularly relevant in the case of this work, since C4 from other geminiviruses has been shown to affect BR signaling, and exogenous application of BR has been shown to alleviate TYLCV-caused symptoms in tomato (Seo et al., 2018) and BCTV C4-induced developmental abnormalities in Arabidopsis (Mills-Lujan and Deom, 2010).

- Please check the spelling of “ethanol” in Figure 9.

- Please follow the ICTV guidelines on how to write virus names.

PLOS authors have the option to publish the peer review history of their article (what does this mean?). If published, this will include your full peer review and any attached files.

Reviewer #1: No

Reviewer #2: No
---

## [Editor Report · Decision Letter 1]

22 Mar 2022

Dear Dr. Qing,

We are pleased to inform you that your manuscript 'Geminivirus C4 proteins inhibit GA signaling via prevention of NbGAI degradation, to promote viral infection and symptom development in N. benthamiana' has been provisionally accepted for publication in PLOS Pathogens.

Best regards,

Aiming Wang, Ph.D

Associate Editor

PLOS Pathogens

Peter Nagy

Section Editor

PLOS Pathogens

Kasturi Haldar

Editor-in-Chief

PLOS Pathogens

orcid.org/0000-0001-5065-158X

Michael Malim

Editor-in-Chief

PLOS Pathogens

orcid.org/0000-0002-7699-2064
---

## [Editor Report · Acceptance letter]

4 Apr 2022

Dear Dr. Qing,

We are delighted to inform you that your manuscript, "Geminivirus C4 proteins inhibit GA signaling via prevention of NbGAI degradation, to promote viral infection and symptom development in N. benthamiana," has been formally accepted for publication in PLOS Pathogens.

Best regards,

Kasturi Haldar

Editor-in-Chief

PLOS Pathogens

orcid.org/0000-0001-5065-158X

Michael Malim

Editor-in-Chief

PLOS Pathogens

orcid.org/0000-0002-7699-2064